# Effect of Fermented Fish Oil on Fine Particulate Matter-Induced Skin Aging

**DOI:** 10.3390/md17010061

**Published:** 2019-01-18

**Authors:** Yu Jae Hyun, Mei Jing Piao, Kyoung Ah Kang, Ao Xuan Zhen, Pincha Devage Sameera Madushan Fernando, Hee Kyoung Kang, Yong Seok Ahn, Jin Won Hyun

**Affiliations:** 1School of Medicine, Jeju National University, Jeju 63243, Korea; yujae_1113@naver.com (Y.J.H.); meijing0219@hotmail.com (M.J.P.); legna48@hanmail.net (K.A.K.); zhenaoxuan705@gmail.com (A.X.Z.); sameeramadhu91@gmail.com (P.D.S.M.F.); pharmkhk@jejunu.ac.kr (H.K.K.); 2Choung Ryong Fisheries Co. LTD, 7825 Iljudong-ro, Namwon-epu, Seogwipo, Jeju 63612, Korea; ecoil@daum.net

**Keywords:** particulate matter, matrix metalloproteinase, fermented fish oil, oxidative stress, skin aging

## Abstract

Skin is exposed to various harmful environmental factors such as air pollution, which includes different types of particulate matter (PM). Atmospheric PM has harmful effects on humans through increasing the generation of reactive oxygen species (ROS), which have been reported to promote skin aging via the induction of matrix metalloproteinases (MMPs), which in turn can cause the degradation of collagen. In this study, we investigated the effect of fermented fish oil (FFO) derived from mackerel on fine PM (particles with a diameter < 2.5 µm: PM_2.5_)-induced skin aging in human keratinocytes. We found that FFO inhibited the PM_2.5_-induced generation of intracellular ROS and MMPs, including MMP-1, MMP-2, and MMP-9. In addition, FFO significantly abrogated the elevation of intracellular Ca^2+^ levels in PM_2.5_-treated cells and was also found to block the PM_2.5_-induced mitogen-activated protein kinase/activator protein 1 (MAPK/AP-1) pathway. In conclusion, FFO has an anti-aging effect on PM_2.5_-induced aging in human keratinocytes.

## 1. Introduction

Reactive oxygen species (ROS) have unpaired electrons and unstable bonds, which are properties that can lead to cellular damage [1,2] and the regulation transcription factors such as AP-1 and NF-kB [3,4]. Cellular ROS can accumulate in response to exogenous stimuli, including air pollution [4].

The skin is the largest organ in the human body and acts as the first-line defense barrier against harmful stimuli such as ultraviolet (UV) light and air pollution, including particulate matter (PM). Depending on particle size, PM can be classified as ultrafine (particles with a diameter < 0.1 µm: PM_0.1_), fine (particles with a diameter < 2.5 µm: PM_2.5_), and coarse (particles with a diameter < 10 µm: PM_10_) [5]. PM can lead to the development of various skin diseases, including skin aging, alopecia, and skin cancer, through the induction of oxidative stress [6,7]. In addition, PM-induced oxidative stress promoted via the production of ROS and subsequent increase in the activity of matrix metalloproteinases (MMPs) [6,8], including MMP-1, MMP-2, and MMP-9, can cause skin aging as a consequence of the degradation of collagen [9,10].

The generation of ROS has been reported to affect skin aging by increasing the expression of MMP-1 in keratinocytes [11]. Several studies have reported that UVB-induced ROS causes skin photoaging via the promotion of the activity of MMP-1 in human keratinocytes and dermal fibroblasts [12,13], and accordingly, it would be desirable to identify an effective antioxidant that could be used to prevent skin aging.

Oxidative stress is also known to stimulate mitogen-activated protein kinase (MAPK) signaling pathways, which affect the regulation mediated by transcription factor activator protein 1 (AP-1) [14]. Activated (phosphorylated) c-Jun and c-Fos can comprise homodimers or heterodimers that bind to AP-1 binding sites in the promoter region of target genes to promote gene transcription [15] such as that of MMPs [10].

Although previous studies have reported that fermented fish oil (FFO) has an antioxidant and protective effect against UVB-induced oxidative damage [16], the effects of FFO on PM_2.5_-induced skin aging are poorly understood. Therefore, in this study, to gain further insight into the antioxidative properties of FFO we investigated the effects of this oil on PM_2.5_-induced skin aging.

## 2. Results

### 2.1. Effect of FFO on PM_2.5_-Induced Intracellular ROS

Given that ROS has been reported to affect skin aging via the generation of MMPs [11], we measured the generation of intracellular ROS using 2’,7’-dichlorodihydrofluorescein diacetate (DCF-DA) fluorescence dye. The flow cytometry data indicated that PM_2.5_-treated cells induced ROS and that pretreatment with FFO reduced PM_2.5_-induced ROS in HaCaT cells, which was confirmed by confocal microscopy (Figure 1a,b).

### 2.2. PM_2.5_-Induced Keratinocyte Senescence

Next, we measured β-galactosidase activity, a marker of cellular senescence [17], to detect HaCaT cell senescence using flow cytometry and confocal microscopy after SPiDER-βGal staining. PM_2.5_-treated cells showed increased β-galactosidase activity in the cytosol, whereas FFO treatment was found to decrease the PM_2.5_-induced β-galactosidase activity (Figure 2a,b).

### 2.3. Effect of FFO on PM_2.5_-Induced MMP-1 Activation and MMP Expression

In this study, we examined cellular MMP-1 activity, as MMP-1 is known to be primarily responsible for skin aging via collagen degradation [18]. The treatment of cells with PM_2.5_ significantly increased the activation of MMP-1 after 6, 12, and 24 h (Figure 3a), whereas pretreatment with FFO decreased the PM_2.5_-induced activation of MMP-1 (Figure 3b). The expression of MMP-1 mRNA and protein was also increased in PM_2.5_-treated cells, whereas it was decreased in the cells pretreated with FFO (Figure 3c,d). As MMP-2 and MMP-9 have also been reported to be involved in skin aging through the degradation of collagen [19], we analyzed the expression levels of MMP-2 and MMP-9 protein and found that these were also increased by PM_2.5_ treatment and decreased by FFO pretreatment (Figure 3e).

### 2.4. Effect of FFO on PM_2.5_-Induced MAPKs and Intracellular Ca^2+^ Levels

MAPKs, which enhance the expression of MMP-1, are activated by an increase in the intracellular Ca^2+^ levels [20]. Our western blot analysis indicated that PM_2.5_ treatment induced the activation (phosphorylation) of extracellular signal-regulated kinase (ERK) and c-Jun N-terminal kinase (JNK), whereas this activation was decreased following FFO pretreatment. Furthermore, PM_2.5_ treatment induced the activation (phosphorylation) of MAPK kinase (MEK) 1/2 and SAPK/ERK kinase (SEK) 1 (Figure 4a), and significantly increased the levels of intracellular Ca^2+^, whereas FFO decreased the PM_2.5_-induced increase in Ca^2+^ levels (Figure 4b,c).

### 2.5. Effect of FFO on PM_2.5_-Induced Transcription Factor Activator Protein 1 (AP-1) Expression

The nuclear transcription factor, AP-1, regulated by MAPKs, increases the expression of MMPs [12,21]. The activation of MAPKs results in the heterodimerization of c-Jun/c-Fos and the formation of an AP-1 complex [10]. As shown in Figure 4, PM_2.5_ treatment increased MAPK activity and intracellular Ca^2+^ levels. Western blot analysis of c-Fos and phospho-c-Jun levels indicated that FFO significantly decreased the PM_2.5_-induced increase in phospho-c-Jun and c-Fos levels (Figure 5a). In addition, FFO reduced the PM_2.5_-induced AP-1 binding to the MMP-1 promoter (Figure 5b).

## 3. Discussion

The skin is the largest organ in the human body and acts as a first-line defense barrier against harmful environmental stimuli, including PM. In several studies, PM_2.5_ has been reported to have harmful effects associated with inflammatory skin diseases, skin aging, and damage to the respiratory system through the generation of intracellular ROS [6,22,23]. The accumulation of ROS has been reported to induce skin aging through the expression of MMPs, including MMP-1, MMP-2, and MMP-9 [24]. Accordingly, it is desirable to identify an effective antioxidant that could be used to prevent skin aging.

Previous studies have demonstrated that mackerel-derived FFO has a direct ROS scavenging effect and a protective effect against UVB-induced oxidative damage [16]. In the present study, we focused on the effect of mackerel-derived FFO against PM_2.5_-induced skin aging.

We initially measured the generation of intracellular ROS induced by PM_2.5_ treatment, and accordingly found that pretreatment with FFO significantly reduced the induction of intracellular ROS by PM_2.5_ treatment (Figure 1a,b). Given that the generation of ROS causes skin aging, we investigated keratinocyte senescence by examining the activity of the marker enzyme β-galactosidase [25]. We found that PM_2.5_ treatment increased β-galactosidase activity, whereas FFO pretreatment significantly reduced the PM_2.5_-induced β-galactosidase activity (Figure 2a,b). As MMPs are known to cause skin aging through the degradation of collagen [9,10], we also examined the activity of MMP-1 and the expression levels of MMP-1, MMP-2, and MMP-9. Again, we found that FFO has an inhibitory effect on PM_2.5_-induced MMPs (Figure 3). These results indicate that PM_2.5_ induces skin aging by promoting the expression of MMPs, whereas FFO has anti-aging effects via its ROS scavenging activity.

ROS activated MAPK signaling pathways, and the activation of MAPKs, in turn, induced various transcription factors, including AP-1 and NF-κB [14,26]. As a result of the translocation of activated AP-1, heterodimers comprising c-Jun and c-Fos and MMPs were synthesized [27,28]. As shown in Figure 4a, ERK and JNK were activated by PM_2.5_ treatment, and this activation was reduced by FFO pretreatment. The activation of MEK and SEK, upstream of ERK and JNK, respectively, was also increased by PM_2.5_ treatment, and this increase was again decreased by FFO pretreatment (Figure 4a). In addition, intracellular Ca^2+^ levels, which regulate MAPKs, were significantly increased in PM_2.5_-treated cells, whereas pretreatment with FFO was found to decrease the observed PM_2.5_-induced increase in Ca^2+^ levels (Figure 4b,c). Furthermore, phospho-c-Jun and c-Fos levels were increased in PM_2.5_-treated cells and decreased in cells receiving pretreatment with FFO (Figure 5a). We also found that PM_2.5_-induced AP-1 binding to the MMP-1 promoter was reduced by FFO pretreatment (Figure 5b). These results therefore provide evidence that FFO can block the activation of the MAPK/AP-1 pathway induced by PM_2.5_ in human keratinocytes. FFO showed an antioxidant effect in this study and in our previous study [16]. However, we did not assess whether FFO may induce the nuclear factor-erythroid 2-related factor 2-mediated antioxidant enzyme pathway to scavenge ROS. This will be evaluated in a future study.

In conclusion, mackerel-derived FFO has anti-aging effects against PM_2.5_-induced skin aging through a reduction in the generation of intracellular ROS and expression of MMPs (Figure 5c).

## 4. Materials and Methods

### 4.1. Cell Culture and Treatment

HaCaT human keratinocytes (CLS Cell Lines Service GmbH, Eppelheim, Germany) were cultured in DMEM medium (Gibco, Life Technologies Co., Grand Island, NY, USA) supplemented with 10% fetal bovine serum and antibiotics (100 units/mL penicillin, 100 µg/mL streptomycin, and 0.25 µg/mL amphotericin B) (Gibco, Life Technologies Co., Grand Island, NY, USA) at 37 °C in an incubator with a humidified atmosphere of 5% CO_2_. Cells were treated with 50 μg/mL diesel particulate matter NIST 1650b (PM_2.5_) (Sigma-Aldrich Chemical Company, St. Louis, MO, USA) and 20 μg/mL FFO. The preparation of PM_2.5_ and FFO has been described in previous studies [16,29].

### 4.2. Detection of Intracellular ROS

For the detection of intracellular ROS in HaCaT cells, the cells were seeded in 6-well culture plates at a density of 1.0 × 10^5^ cells/well, cultured for 16 h, and treated with 20 μg/mL FFO and 50 μg/mL PM_2.5_. After 30 min, 50 μM 2’,7’-dichlorodihydrofluorescein diacetate (DCF-DA, Molecular Proves, Eugene, OR, USA) solution was added to the cells. After 15 min incubation at 37 °C, DCF fluorescence was measured using a BD LSRFortessa flow cytometer (PerkinElmer, Waltham, MA, USA). For imaging, cells were seeded in a 4-well glass chamber slide at a density of 1.0 × 10^5^ cells/well, cultured for 16 h, and treated with 20 μg/mL FFO and 50 μg/mL PM_2.5_. After 30 min, 50 μM DCF-DA (Molecular Proves, Eugen, OR, USA) solution was added to the cells. After 15 min incubation at 37 °C, the cells were washed 3 times with PBS. DCF fluorescence images were obtained using a FV1200 laser scanning confocal microscope (Olympus, Tokyo, Japan).

### 4.3. Detection of β-Galactosidase Activity

β-galactosidase activity is a classic marker of cellular aging [30], and we accordingly measured the activity of this enzyme to detect cell senescence. Cells were seeded in 6-well culture plates at a density of 1.0 × 10^5^ cells/mL, and after a 16-h incubation at 37 °C, they were treated with 20 μg/mL FFO and 50 μg/mL of PM_2.5_, followed by the addition of 2 μM SPiDER-βGal solution (Dojindo Molecular Technologies, Inc., Rockville, MD, USA) 24 h later. After a 15-min incubation at 37 °C, SPiDER-βGal fluorescence was measured using a BD LSRFortessa flow cytometer (PerkinElmer, Waltham, MA, USA). For imaging, cells were seeded in a 4-well glass chamber slide at a density of 1.0 × 10^5^ cells/well, and after a 16 h incubation at 37 °C, they were treated with 20 μg/mL FFO and 50 μg/mL of PM_2.5_, followed by the addition of 2 μM SPiDER-βGal solution (Dojindo Molecular Technologies, Inc., Rockville, MD, USA) 24 h later. After a 15 min incubation at 37 °C, the cells were washed 3 times with PBS. After washing, the cells were treated with mounting medium containing DAPI to label nuclei. SPiDER-βGal fluorescence images were obtained using an FV1200 laser scanning confocal microscope (Olympus, Tokyo, Japan).

### 4.4. MMP-1 Activity

MMP-1 activity was measured using a Fluorokine^®^ E human active MMP-1 fluorescent assay kit (R&D Systems Inc., Minneapolis, MN, USA). HaCaT cells were seeded in a 60 mm culture dish at 1.0 × 10^5^ cells/mL. After a 16 h incubation at 37 °C, the cells were treated with 20 μg/mL FFO, and 1 h thereafter with 50 μg/mL PM_2.5_. MMP-1 activity was assessed according to the manufacturer’s instructions. Fluorescence was measured using a Spectra Max i3x microplate reader (Molecular Devices, San Jose, CA, USA).

### 4.5. Reverse Transcription–PCR (RT-PCR)

Cells were seeded in a 60 mm culture dish at 1.5 × 10^5^ cells/mL, and after a 1 h incubation at 37 °C, cells were treated with 20 μg/mL FFO and 50 μg/mL PM_2.5_. After 24 h, we isolated total RNA from cells using an easy-BLUE™ total RNA extraction kit (iNtRON Biotechnology Inc., Seongnamsi, Korea). Subsequently, cDNA was amplified using reverse transcription reaction buffer, primers, dNTPs, and Taq DNA polymerase in a final volume of 20 μL. The amplified products were mixed with 6× blue/orange loading dye and resolved by electrophoresis on 1% agarose gels. Gels were subsequently stained with RedSafe™ nucleic acid staining solution (iNtRON Biotechnology Inc., Seongnamsi, Korea), photographed under a UV light, and analyzed using Image Quant™ TL analysis software (Amersham Biosciences, Uppsala, Sweden). The PCR conditions used were as follows: an initial denaturation at 94 °C for 5 min, followed by 30 cycles of 94 °C for 30 s, 55 °C for 30 s, and 72 °C for 1 min. The primers used in this study were as follows: human MMP-1: forward (5’-GGAGGAAATCTTGCTCAT-3’) and reverse (5’-CTCAGAAAGAGCAGCATC-3’); human GAPDH: forward (5′-TCAAGTGGGGCGATGCTGGC-3’) and reverse (5′-TGCCAGCCCCAGCGTCAAAG-3’).

### 4.6. Western Blot Analysis

Protein lysates (30 μg per lane) were electrophoresed on 12% SDS-polyacrylamide gels, and then transferred to nitrocellulose membranes, which were incubated with the primary antibodies and subsequently with HRP-conjugated secondary antibodies (Invitrogen, Carlsbad, CA, USA). Thereafter, the membranes were developed using a western blotting detection kit (GE Healthcare Life Sciences, Buckinghamshire, Little Chalfont, UK) for detection of the protein bands and then exposed to X-ray film. The primary antibodies used in this study were as follows: MMP-1 (Cusabio Technology LLC., Houston, TX, USA), MMP-2 (Abcam, Cambridge, UK), MMP-9 (Abcam, Cambridge, UK), phospho-c-Jun (Cell Signaling Technology, Danvers, MA, USA), c-Fos (Cell Signaling Technology, Danvers, MA, USA), phospho-SEK (Cell Signaling Technology, Danvers, MA, USA), phospho-MEK (Cell signaling Technology, Danvers, MA, USA), phospho-ERK (Santa Cruz Biotechnology, Santa Cruz, CA, USA), phospho-JNK (Cell Signaling Technology, Danvers, MA, USA), and Actin (Sigma-Aldrich Chemical Company, St. Louis, MO, USA).

### 4.7. Measurement of Ca^2+^ Levels

To determine cellular Ca^2+^ levels, cells were seeded in 6-well culture plates at a density of 1.0 × 10^5^ cells/well, cultured for 16 h, and then treated with 20 μg/mL FFO and 50 μg/mL PM_2.5_. After 24 h, 5 μM Fluo-4-AM (Molecular Probes, Eugene, OR, USA) solution was added to the cells. After 15 min incubation at 37 °C, Fluo-4-AM fluorescence was measured using a BD LSRFortessa flow cytometer (PerkinElmer, Waltham, MA, USA). For imaging, cells were seeded in a 4-well glass chamber slide at a density of 1.0 × 10^5^ cells/well, cultured for 16 h, and then treated with 20 μg/mL FFO and 50 μg/mL PM_2.5_. After 24 h, 5 μM Fluo-4-AM (Molecular Probes, Eugen, OR, USA) solution was added to the cells. After a 15 min incubation at 37 °C, the cells were washed 3 times with PBS. After washing, Flou-4-AM fluorescence images were obtained using an FV1200 laser scanning confocal microscope (Olympus, Tokyo, Japan).

### 4.8. Chromatin Immunoprecipitation (ChIP) Assay

ChIP assays were performed using a SimpleChIP™ enzymatic chromatin IP kit (Cell Signaling Technology, Danvers, MA, USA). HaCaT cells were seeded in a 100 mm culture dish at 1.5 × 10^5^ cells/mL, and after 16 h, were treated with 20 μg/mL FFO and 50 μg/mL PM_2.5_. All processes were performed according to the manufacturer’s instructions. As an antibody, we used a c-Jun antibody (Invitrogen, Carlsbad, CA, USA). The following primers were designed from the MMP-1 gene promoter (−67 to +94 of the MMP-1 gene sequence from the transcription starting site; Bionics): sense 5′-CCTCTTGCTGCTCCAATATC-3′ and antisense 5′-TCTGCTAGGAGTCACCATTTC-3′. PCR products were separated on 1% agarose gels. DNA bands were photographed under a UV light and were analyzed using Image Quant™ TL analysis software version 1.0.2 (Amersham Biosciences, Uppsala, Sweden).

### 4.9. Statistical Analysis

All analyses were performed in triplicate and all values are expressed as the mean ± standard error of the means. The results were assessed using one-way analysis of variance with Tukey’s test using the SigmaStat version 3.5 software (Systat Software Inc., San Jose, CA, USA) to determine the statistical significance of differences between means. A *p*-value < 0.05 was considered to be statistically significant.

## Figures and Tables

**Figure 1 marinedrugs-17-00061-f001:**
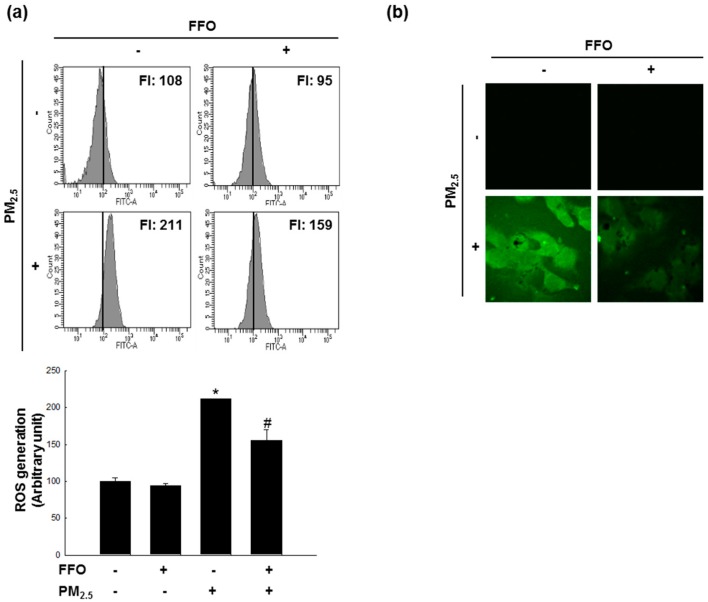
Scavenging effect of fermented fish oil (FFO) on PM_2.5_-induced intracellular reactive oxygen species (ROS). Intracellular ROS were detected by (**a**) flow cytometry and (**b**) confocal microscopy after DCF-DA staining. * *p* < 0.05 and # *p* < 0.05 compared with untreated cells and PM_2.5_-treated cells, respectively.

**Figure 2 marinedrugs-17-00061-f002:**
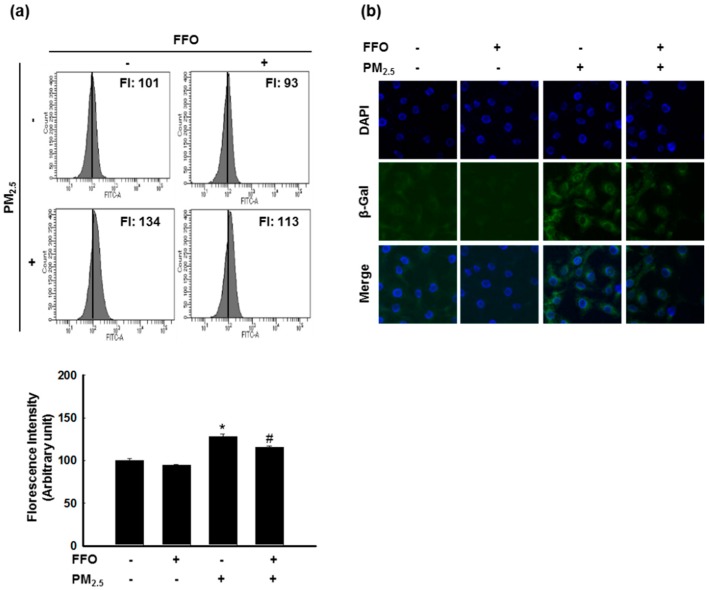
PM_2.5_-induced keratinocyte senescence. (**a**) β-galactosidase activity was measured by flow cytometry and (**b**) confocal microscopy after SPiDER-βGal staining. * *p* < 0.05 and # *p* < 0.05 compared with untreated cells and PM_2.5_-treated cells, respectively.

**Figure 3 marinedrugs-17-00061-f003:**
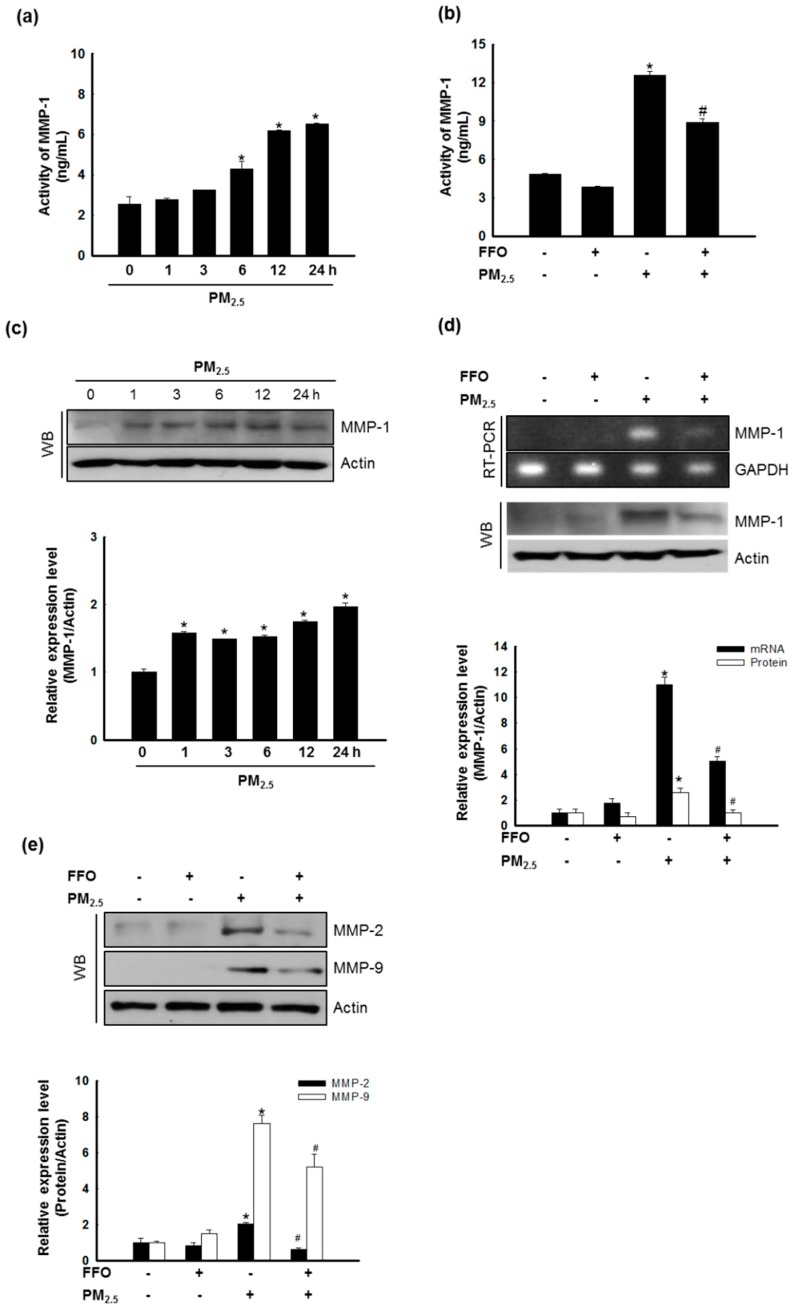
Effect of fermented fish oil (FFO) on PM_2.5_-induced MMP-1 activation and matrix metalloproteinase (MMP) expression. (**a**) The MMP-1 activity of PM_2.5_-treated cells at the indicated times and (**b**) the MMP-1 activity of FFO and PM_2.5_ treated cells were determined using a human active MMP-1 fluorescent assay kit. * *p* < 0.05 and # *p* < 0.05 compared with untreated cells and PM_2.5_-treated cells, respectively. (**c**) Expression level of MMP-1 was analyzed by western blot analysis. Actin was used as a loading control. * *p* < 0.05 compared with untreated cells. (**d**) The mRNA and protein levels of MMP-1 were analyzed by RT-PCR and western blotting, respectively. GAPDH and actin were used as loading controls. * *p* < 0.05 and # *p* < 0.05 compared with untreated cells and PM_2.5_-treated cells, respectively. (**e**) Expression level of MMP-2 and MMP-9 were analyzed by western blotting. Actin was used as a loading control. * *p* < 0.05 and # *p* < 0.05 compared with untreated cells and PM_2.5_-treated cells, respectively.

**Figure 4 marinedrugs-17-00061-f004:**
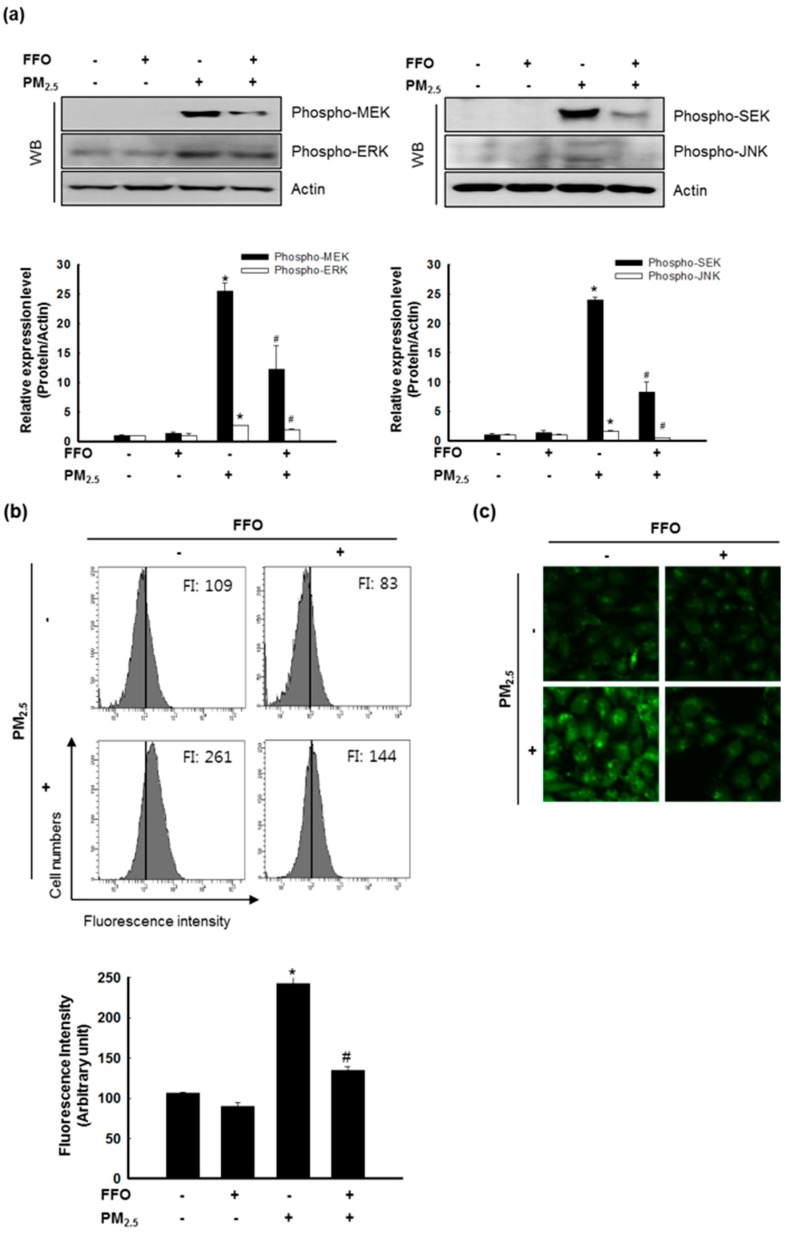
Effect of fermented fish oil (FFO) on PM_2.5_-induced MAPKs and intracellular Ca^2+^ level. (**a**) Expression level of phospho-MEK, phospho-ERK, phospho-SEK, and phospho-JNK determined by western blot analysis. Actin was used as a loading control. * *p* < 0.05 and # *p* < 0.05 compared with untreated cells and PM_2.5_-treated cells, respectively. (**b**) Intracellular Ca^2+^ level was detected by flow cytometry and (**c**) confocal microscopy after Flou-4-AM staining. * *p* < 0.05 and # *p* < 0.05 compared with untreated cells and PM_2.5_-treated cells, respectively.

**Figure 5 marinedrugs-17-00061-f005:**
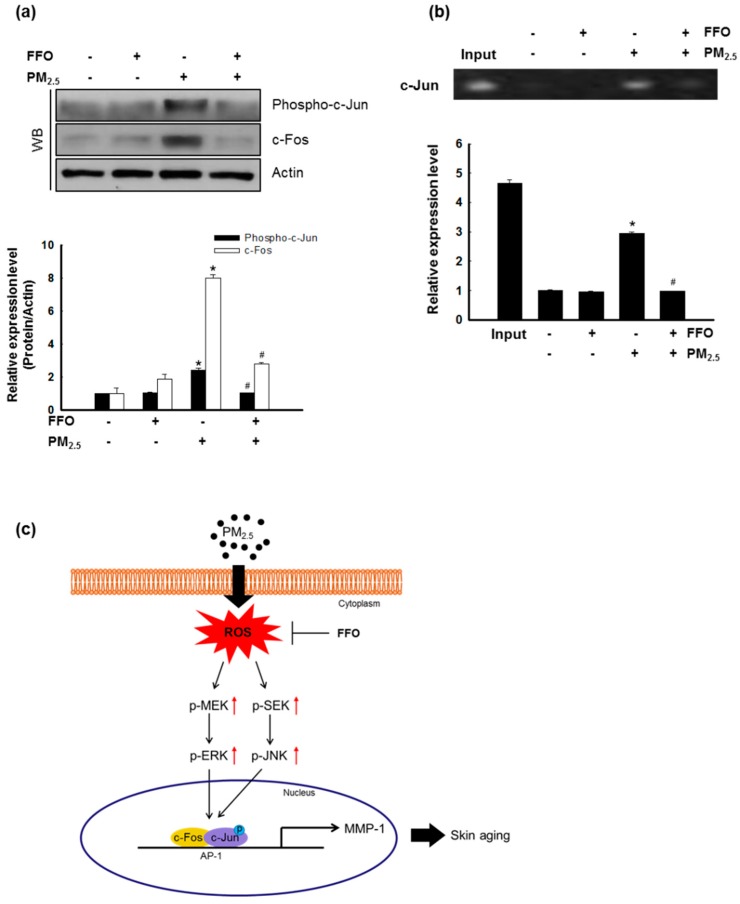
Effect of fermented fish oil (FFO) on PM_2.5_-induced transcription factor activator protein 1 (AP-1) expression. (**a**) Expression level of phospho-c-Jun and c-Fos determined by western blot analysis. Actin was used as a loading control. * *p* < 0.05 and # *p* < 0.05 compared with untreated cells and PM_2.5_-treated cells, respectively. (**b**) AP-1 binding to the MMP-1 promoter was assessed by ChIP assay. * *p* < 0.05 and # *p* < 0.05 compared with untreated cells and PM_2.5_-treated cells, respectively. (**c**) Schematic diagram of the effect of FFO on PM_2.5_-induced skin aging. Exposure to PM_2.5_ increased intracellular reactive oxygen species (ROS) and induced skin aging in HaCaT cells. The mackerel-derived FFO had an anti-aging effect on PM_2.5_-induced skin aging through reduced intracellular ROS.

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
