# Peer review of "Effect of Fermented Fish Oil on Fine Particulate Matter-Induced Skin Aging"

_marinedrugs, 2019, doi:10.3390/md17010061_

Round 1
Reviewer 1 Report
The present manuscript is an interesting research of the fermented fish oil effects on ROS induced skin aging.
The results obtained demonstrate the reduction of ROS by FFO in human keratinocytes after treatment with PM2,5 particles, as well as the inactivation of MMP-1, MMP-2 and MMP-9, in addition to the inhibition of the MAPKs signalling cascades, and the MMP-1 promoter activation by AP-1.
Regarding the presentation of those results obtained, Figure 1b do not show a clear reduction of ROS levels by confocal microscopy. Authors would need to replace these images in order to show equivalent results to those determined by flow cytometry.
In Figure 4a, the western blot results for Phospho-JNK and their quantification are not so evident. It would be desirable to show better WB gel images where those effects could be appreciated clearly.
In the Materials and Methods part of the manuscript, the authors need to describe in a more detailed way the format of the cell plates used (P6, P24, P48 or P96) in each experiment, the time of incubation and the media with fluorescence probes, cell washing procedures, etc.
With regard to the Statistical Analysis, more information in needed about the methods employed, i.e. One Way Anova, and the statistical package used.
Author Response
Reviewer 1
The present manuscript is an interesting research of the fermented fish oil effects on ROS induced skin aging. The results obtained demonstrate the reduction of ROS by FFO in human keratinocytes after treatment with PM2,5 particles, as well as the inactivation of MMP-1, MMP-2 and MMP-9, in addition to the inhibition of the MAPKs signalling cascades, and the MMP-1 promoter activation by AP-1.
Regarding the presentation of those results obtained, Figure 1b do not show a clear reduction of ROS levels by confocal microscopy. Authors would need to replace these images in order to show equivalent results to those determined by flow cytometry.
Response: As suggested by the reviewer, Figure 1b has been replaced with a better image.
In Figure 4a, the western blot results for Phospho-JNK and their quantification are not so evident. It would be desirable to show better WB gel images where those effects could be appreciated clearly.
Response: We have changed the phospho-JNK image in Figure 4a.
In the Materials and Methods part of the manuscript, the authors need to describe in a more detailed way the format of the cell plates used (P6, P24, P48 or P96) in each experiment, the time of incubation and the media with fluorescence probes, cell washing procedures, etc.
Response: As suggested, we have indicated the detailed procedures in the Materials and
Methods section.
With regard to the Statistical Analysis, more information in needed about the methods employed, i.e. One Way Anova, and the statistical package used.
Response: We have provided more information on the statistical analysis in the Statistical analysis subsection of the revised manuscript.

Reviewer 2 Report
In their manuscript Hyun and coworkers analyze the harmful effect of Atmospheric PM on human immortalized keratinocytes, HaCaT cells. As reported in the literature they observed that PM treatment increased the generation of reactive oxygen species and induction of matrix metalloproteinases (MMPs) that in turn promote skin aging. In their manuscript, the authors investigated the effect of fermented fish oil derived on fine PM-mediated skin aging. They report some clear and interesting observations. They found that FFO inhibited the PM-induced generation of intracellular ROS and MMPs activity. Mechanistically they found that FFO abrogated the elevation of intracellular Ca2+ levels and blocked the PM induced MAPK/AP-1 pathway. The experiments are clear and well performed and deserve publication. Although, behind the scope of the present manuscript, it would be interesting to address, or at least discuss, if in their experimental conditions, the effect on ROS levels of FFO treatment occurs through a NRF2-mediated mechanism.
Author Response
Reviewer 2
In their manuscript Hyun and coworkers analyze the harmful effect of Atmospheric PM on human immortalized keratinocytes, HaCaT cells. As reported in the literature they observed that PM treatment increased the generation of reactive oxygen species and induction of matrix metalloproteinases (MMPs) that in turn promote skin aging. In their manuscript, the authors investigated the effect of fermented fish oil derived on fine PM-mediated skin aging. They report some clear and interesting observations. They found that FFO inhibited the PM-induced generation of intracellular ROS and MMPs activity. Mechanistically they found that FFO abrogated the elevation of intracellular Ca2+ levels and blocked the PM induced MAPK/AP-1 pathway. The experiments are clear and well performed and deserve publication. Although, behind the scope of the present manuscript, it would be interesting to address, or at least discuss, if in their experimental conditions, the effect on ROS levels of FFO treatment occurs through a NRF2-mediated mechanism.
Response: The fermented fish oil (FFO) showed an antioxidant effect in this study and in our previous study [16]. However, we did not assess whether FFO may induce nuclear factor-erythroid 2-related factor 2 (Nrf2)-mediated antioxidant enzyme pathway to scavenge ROS. This will be evaluated in a further study. We have indicated this in the Discussion section of the revised manuscript.
